# Combined Analysis of the Whole Transcriptome of Piglets Infected with SADS−CoV Virulent and Avirulent Strains

**DOI:** 10.3390/microorganisms11020409

**Published:** 2023-02-06

**Authors:** Qianniu Li, Xiaoyu Tang, Ling Zhou, Xiaocheng Lv, Long Gao, Tian Lan, Yuan Sun, Jingyun Ma

**Affiliations:** 1Guangdong Provincial Key Laboratory of Agro−Animal Genomics and Molecular Breeding, College of Animal Science, South China Agricultural University, Guangzhou 510642, China; 2Guangdong Laboratory for Lingnan Modern Agriculture, Guangzhou 510642, China

**Keywords:** SADS−CoV, RNA, avirulent strains, virulent strains, HAllA

## Abstract

When piglets are infected by virulent and avirulent strains of swine acute diarrhea syndrome coronavirus (SADS−CoV), there are obvious differences in their clinical symptoms; however, the specific mechanisms of pathogenicity and the immune regulation of highly pathogenic and low pathogenic strains are unknown. We collected intestinal tissues from SADS−CoV−infected piglets, performed a whole transcriptome sequencing analysis, including mRNA, miRNA, lncRNA, cicrRNA, and TUCP, and performed functional and correlation analyses of differentially expressed RNAs. Our results showed that the differentially expressed RNAs in group A versus group B (AvsB), group A versus group C (AvsC), and group B versus group C (BvsC) were relevant to immune and disease−related signaling pathways that participate in the organisms’ viral infection and immune regulation. Furthermore, data obtained from the HAllA analysis suggested that there was a strong correlation between the differentially expressed RNAs. Specifically, LNC_011487 in the P set was significantly negatively correlated with ssc−miR−215, and LNC_011487 was positively correlated with PI3. Moreover, we also constructed a differentially expressed RNA association network map. This study provides a valuable resource for studying the SADS−CoV transcriptome and pathogenic mechanism from the perspective of RNA to understand the differences in and consistency of the interaction between virulent and attenuated SADS−CoV strains and hosts.

## 1. Introduction 

Swine acute diarrhea syndrome coronavirus (SADS−COV) is a newly discovered porcine coronavirus, that was first reported in China in 2017 [1]. Accordingly, SADS−CoV can cause swine acute diarrhea syndrome in piglets. Its clinical outcomes include acute diarrhea, vomiting, rapid weight loss in piglets less than one week old, and high mortality (more than 90%) [2,3]. SADS−CoV is a single−stranded, positive−stranded envelope virus, belonging to the genus *Alphacoronavirus* of the family *Coronaviridae* [1]. The diameter of SADS−CoV virus particles is about 100 nm. Moreover, filamentous protrusions similar to those of the coronavirus have been found on the surface of SADS−CoV’s envelope. The SADS−CoV genome length is approximately 27 kb, and its genome encodes five non−structural proteins, ORF1a, ORF1b, NS3a, NS7a, and NS7b, and four structural proteins, S, E, M, and N [1,3].

There is a complex relationship between a virus and its host. Hence, understanding this relationship can help uncover the underlying mechanism of virus replication and infection in hosts and also is beneficial for developing disease prevention and control strategies. Viruses have evolved to manipulate multiple signal transduction pathways in favor of their own reproduction. It has been recorded that the SADS−CoV N protein could antagonize IFN−β production by inducing RIG−I ubiquitination [4]. Furthermore, our early study found that SADS−CoV could inhibit IFN−β production by targeting IPS−1 [5]. Mechanistically, the SADS−CoV N protein is an IFN−β antagonist, and it inhibits IFN−β production by targeting TBK1 to interfere with the interaction between TRAF3 and TBK1 [6]. Zhang et al. found that the inhibition of ZERK activity was associated with reduced apoptosis in SADS−CoV−infected cells, suggesting a critical role for ERK signaling in the SADS−CoV life cycle [7]. Edwards et al. found that rSADS−CoV did not use human coronavirus ACE−2, DPP4, or CD13 receptors for docking or entry [8]. At present, there are few studies about the mechanism of SADS−CoV infection, or hosts and their potential receptors. Therefore, studying the interaction mechanism between a virus and its host is particularly pivotal.

Biosafety precautions must be taken against the virus, as there is no vaccine for its treatment or defense. Only one study has stated that ZDHHC17 may be a potential drug target for SADS−CoV infection [9]. However, the pathogenic mechanism of SADS−CoV is still unclear. In this context, our laboratory created an attenuated SADS−CoV strain by serial passage. Moreover, virulent and avirulent strains were employed for animal challenge experiments. After attacks by avirulent strains, the tissue lesions, anal swab detoxification, and tissue tropism were relatively weakened [10]. To better understand the relationship between virulent strains and weak virulent strains, a whole transcriptome analysis, including mRNA, miRNA, lncRNA, cicrRNA, and TUCP, was performed, and the data were subjected to a basic bioinformatics analysis as well as expression/abundance detection. The results of differential expression (*p* ≤ 0.05) were obtained after a differential expression analysis between the samples, from which the results of significant differences were filtered out, and the molecular events related to pathogenicity (P set) and immune response (I–V set) were detected. The results elucidate the difference and consistency of the interaction between the strong and weak strains of SADS−CoV and hosts from the perspective of RNA, allow us to study its pathogenic mechanism, and provide reference and support for the development of a vaccine. 

## 2. Materials and Methods

### 2.1. Virus Strain 

The passaged strains SADS−CoV−CN/GDWT/2017−P7 and SADS−CoV−CN/GDWT/2017−P83 of the SADS−CoV−CN/GDWT/2017 strain were isolated and preserved by our laboratory. Serum−free DMEM contained 10 μg/mL trypsin (Invitrogen, CA, USA) Infected cells were incubated at 37 °C in an incubator containing 5% CO_2_ and monitored daily for cytopathic effects (CPEs). When 90% of cells showed CPEs, the cells were harvested, frozen and thawed three times, and centrifuged at 8000× *g* for ten minutes. The supernatant was collected and stored at −80 °C as a stock solution for the next passaging. Using the same approach, 82 subsequent passages were performed in Vero cells. They were purified every ten generations, starting from the 8th generation of cell culture. Viral titers were determined by TCID50 every five or ten generations. 

### 2.2. Piglet Challenge Experiments

The piglets were purchased from a farm owned by Huanong Wen’s Co., Ltd. In the experiment, we selected piglets weighing around two kilograms, which were responsive and energetic, had a good appetite, and were sensitive to external stimuli. They were divided into three groups: A (high pathogenicity group), B (low pathogenicity group), and C (control group) after ear tags were added. Piglets were housed in separate animal rooms to ensure no cross−contamination. Anal swabs were collected before the challenge. No specific virus infection was detected by RT−PCR, and SADS−CoV, porcine deltacoronavirus (PDCoV), transmissible gastroenteritis virus (TGEV), porcine epidemic diarrhea virus (PEDV), porcine rotavirus (RV), and other diarrhea pathogenic viruses were not detected. The piglets were sent to the animal room when they were 5 days old, and the challenge test was carried out after they had adapted to their environment. Group A was infected with 12 mL of virulent strain (P7) solution containing 1.0  × 10^6.3^ TCID50/mL administered orally (9 piglets), group B was infected with 12 mL of avirulent strain (P83) solution containing 1.0 × 10^6.3^ TCID 50/mL administered orally (9 piglets), and 8 control piglets were administered the same volume of DMEM.

The peak incidence period was from day 3 to day 5 post−challenge. Per group, three piglets were randomly selected to be bled and slaughtered, and the intestines were exposed after dissection of the piglets’ abdominal cavity, followed by sampling of the intestines. The intestine was the ileum of the small intestine tissue, and 0.2 g was collected per sample. Intestinal tissues were lysed by grinding after we added TRIZOL reagent (Invitrogen, CA, USA). The tissue was fully ground, and then freeze−thawed three times for nucleic acid extraction.

### 2.3. Library Construction and Sequencing Process

Firstly, the quality of the sample was tested, the RNA integrity of the sample and the presence of DNA contamination were analyzed, and the RNA purity (OD260/280 and OD260/230 ratio) and the accurate detection of RNA integrity were examined [11]. After the sample was qualified, the Small RNA Sample Pre Kit (Illumina, San Diego, CA) was used to construct the library, which is designed to generate small RNA libraries directly from total RNA. Using the special structure of the 3’ and 5’ ends of the small RNA and with the total RNA as the starting sample, we directly added the adapters to both ends of the small RNA. After that, the cDNA synthesis was performed using kit. After PCR amplification, PAGE gel electrophoresis was conducted to separate the target DNA fragments, and then recovered cDNA library was obtained by cutting the gel [12]. The LncRNA/TUCP library uses strand−specific library construction [13]. We used the method of reverse transcription to synthesize the first strand of cDNA for the general library construction method of NEB with slight modulation. In contrast, when the second strand was synthesized, dTTP in dNTPs was replaced by dUTP, followed by cDNA end repair, A−tailing, ligation of sequencing adapters, and length screening [14,15]. After utilizing the USER enzyme to degrade the second strand of cDNA containing U, PCR amplification was conducted to obtain the library [16]. After the library was constructed, Qubit2.0 was used for preliminary quantification, then the library was diluted to 1 ng/L, and Agilent 2100 was used to detect the insert size of the library. After the insert size met our expectations, the effective concentration of the library was quantified by qPCR method (the effective concentration of the library was >2 nM) to ensure the quality of the library. After that, the different libraries were pooled according to the requirements of effective concentration and target data volume, and then HiSeq/MiSeq sequencing was performed.

### 2.4. Differentially Expressed RNA Screening

The overall distribution of differentially expressed RNAs was inferred by a volcano plot, and the differentially expressed RNAs were screened by evaluating them from the fold change and the corrected significance level (padj/qvalue). The default screening condition for differentially expressed RNA was padj < 0.05. The default differentially expressed RNA screening conditions were qvalue < 0.01 &|log2(fold change)| > 1.

### 2.5. GO and KEGG Enrichment Analysis

The GO enrichment analysis was conducted using GO seq, based on Wallenius’ non−central hypergeometric distribution, compared with the ordinary hypergeometric distribution, the characteristic of which is that the probability of extracting an individual from a certain category is different from the probability of extracting an individual from outside a certain category. This difference in probability is obtained by estimating the preference of gene length, so that the probability of GO term being enriched by candidate target genes can be calculated more accurately [17].

In organisms, different genes coordinate with each other to perform their biological functions. Pathway significant enrichment can determine the most important biochemical metabolic pathways and signal transduction pathways involved in candidate target genes. The KEGG (Kyoto Encyclopedia of Genes and Genomes) is the main public database on pathways [18]. Pathway significant enrichment analysis takes KEGG Pathway as its unit and applies hypergeometric test to find pathways that are significantly enriched in candidate target genes compared with the whole genome background.

### 2.6. Multi−Omics Association Analysis

Hierarchical all−against−all association (HAllA) is a computational method for finding multi−resolution associations in high−dimensional, heterogeneous datasets, and for discovering significant relationships between data features with high power [19]. HAllA has strong robustness of data types, operating on continuous and categorical values, and on homogeneous datasets (all measurements are of the same type, such as gene expression microarrays) and heterogeneous (contains measurements with different units or types, such as patient clinical data) datasets both work well.

We used the HAllA multi−omics association method (R package: halla) to perform multi−omics association analysis according to the following comparison schemes: circRNA vs mRNA, circRNA vs miRNA, lncRNA vs miRNA, miRNA vs mRNA, and lncRNA/TUCP vs mRNA [20]. A, B, and C represent the lethality group, the low−pathogenicity group, and the control group, respectively. The molecular events associated with pathogenicity in AvsC are consistent with those in AvsB. The immune response generated in AvsC contains a response to bacterial infection secondary to SADS−CoV, and the immune response generated in BvsC is purely a host immune response to SADS−CoV. Systematic errors were randomized. Data for each group were used to correlate molecular events related to pathogenicity (the intersection of AvsC and AvsB represents molecular events associated with viral potential pathogenicity, lethality, and inflammatory responses secondary to bacterial infection, P set) and immune response the intersection of BvsC and AvsC represents the host immune response to SADS−CoV, I–V set). The correlation of the correlation results, based on the pairwise Pearson correlation coefficients, were adjusted by false discovery rate (FDR < 0.05).

### 2.7. Construction of Differentially Expressed RNA Association Network

According to the association analysis results of HAllA of P set and I–V set, we used Cytoscape v3.2.1 to build an association network graph.

## 3. Results

### 3.1. Expression Profiles of Differentially Expressed mRNAs

After analyzing the differentially expressed mRNAs in different groups, it was found that there were 1955 differentially expressed mRNAs in the AvsB group, 890 differentially expressed mRNAs in the AvsC group, and 2074 mRNAs in the BvsC group with significant changes (Figure 1A–C). Among them, the expression of SPARCL1 (SPARC like 1) (https://www.genecards.org/cgi−bin/carddisp.pl?gene=SPARCL1&keywords=SPARCL1 accessed on 27 January 2023) was up−regulated 7.9 times in the AvsB group. However, in the other groups, it did not change significantly in expression. The expression of HNRNPH1 (Heterogeneous Nuclear Ribonucleoprotein H1) (https://www.genecards.org/cgi−bin/carddisp.pl?gene=HNRNPH1 accessed on 27 January 2023) was down−regulated 7.2 times. The expression of SLA−1 only in the AvsC group was up−regulated 8.08 times. In the BvsC group, the upregulation of SLC5A1 (Solute Carrier Family 5 Member 1) (https://www.genecards.org/cgi−bin/carddisp.pl?gene=SLC5A1 accessed on 27 January 2023) was the most pronounced. After the GO enrichment analysis, it was found that the mRNA of the AvsB group was mainly related to protein activity and metabolism−related processes. The mRNA of the AvsC group was mainly enriched in immune response and catalysis. In the BvsC group, mRNAs were enriched in protein metabolism−related processes (Figure 1D–F). The results of the KEGG analysis showed that the mRNA was significantly enriched in metabolic pathways in the AvsB group, and the mRNA in AvsC group was significantly enriched in disease−related pathways and antigen processing and presentation (Figure 1G,H). In the BvsC group, mRNAs were significantly enriched in immune−related signaling pathways, such as the Toll−like receptor signaling pathway, NF−kappa B signaling pathway, and T cell receptor signaling pathway (Figure 1I).

The P set had 70 intersection mRNAs, of which TGM3 (Transglutaminase 3) (https://www.genecards.org/Search/Keyword?queryString=TGM3) is related to a virus’s entry into host cells and the virus cycle. FUT2 (Fucosyltransferase 2) (https://www.genecards.org/Search/Keyword?queryString=FUT2 accessed on 27 January 2023), CCL5 (C−C Motif Chemokine Ligand 5) (https://www.genecards.org/Search/Keyword?queryString=CCL5 accessed on 27 January 2023), SLA−DQA1, and CXCL9 (C−X−C Motif Chemokine Ligand 9) (https://www.genecards.org/Search/Keyword?queryString=CXCL9 accessed on 27 January 2023) in the intersection mRNA of the I–V set are all related to immune response, so these intersection mRNAs can be used as the research target of immune response (Appendix A).

### 3.2. Expression Profiles of Differentially Expressed miRNAs

After analyzing the differentially expressed miRNAs in different groups, it was found that there were 66 differentially expressed miRNAs in the AvsB group, 51 differentially expressed miRNAs in the AvsC group, and 8 in the BvsC group. Among them, the expression of ssc−miR−10390 (https://www.mirbase.org/textsearch.shtml?q=ssc−miR−10390 accessed on 27 January 2023) in the AvsB group and AvsC group was down−regulated. In the AvsC group, the expression of ssc−miR−10390 was up−regulated. The expression of the newly discovered miRNA novel_536 was down−regulated in the AvsC group. In the BvsC group, the expression of ssc−miR−21−3p (https://www.mirbase.org/textsearch.shtml?q=ssc−miR−21−3p accessed on 27 January 2023) was significantly down−regulated. After the GO enrichment analysis, it was found that miRNAs in the AvsB group were mainly related to binding, autophagy, and junction processes (Figure 2D). The miRNAs in the AvsC group were mainly enriched in enzyme activator activity, the cytokinetic process, and other processes (Figure 2E). In the BvsC group, miRNAs were enriched in immune−related signaling pathways and phosphors metabolic processes. However, in the GO enrichment analysis of the BvsC group, the corrected *p*−value was >0.05. It is mentioned here for reference only. The KEGG analysis showed that the miRNAs in the AvsB group were significantly enriched in natural−killer−cell−mediated cytotoxicity, the B cell receptor signaling pathway, the NF−kappa B signaling pathway, and other pathways (Figure 2F). It is worth noting that the miRNAs in the AvsB group are significantly enriched in Fc gamma R−mediated phagocytosis and the Fc epsilon RI signaling pathway; all Fc γ receptors belong to the immunoglobulin superfamily; and Fc γ receptors are involved in multiple immune system functions [21]. Similar to the AvsB group, the significantly enriched pathways in the AvsC group were mostly related to immunity (Figure 2G). Like the AvsB group, the BvsC group was also enriched in the Fc gamma R−mediated phagocytosis and Fc epsilon RI signaling pathways (Figure 2H). In addition to this, the most significant pathway enriched in this group was the intestinal immune network for IgA production, which may be related to intestinal lesions after SADS−CoV infection.

The P set had 24 intersection miRNAs, and these miRNAs may be related to molecular events related to the potential pathogenicity, lethality, and inflammatory response of secondary bacterial infection. There were four miRNAs in the intersection of the I–V set, which may have an impact on the immune response of the host to SADS−CoV. Therefore, miRNAs are involved in regulation during SADS−CoV infection (Appendix A).

We analyzed the association of miRNA and mRNA by the HAllA association method, and the results showed that in the pathogenic set (P set), ssc−miR−215 (https://www.mirbase.org/textsearch.shtml?q=ssc−miR−215 accessed on 27 January 2023) had the highest association with the mRNA ABHD3, which was cut into a positive correlation. The second highest correlation was that of ssc−miR−1285 (https://www.mirbase.org/textsearch.shtml?q=ssc−miR−1285 accessed on 27 January 2023) with S100G and SLC30A10. ssc−miR−133a−3p (https://www.mirbase.org/textsearch.shtml?q=ssc−miR−133a−3p accessed on 27 January 2023) was negatively correlated with GSTA1, indicating that these molecules have a high correlation in the P set (Figure 3A). In the immune response collections (IV set), ssc−miR−1388 (https://www.mirbase.org/textsearch.shtml?q=ssc−miR−1388 accessed on 27 January 2023) had the most significant association with LMNA, followed by ssc−miR−1285 and FUT2, both of which were positively correlated, indicating that these molecules were included in the immune response collection. It shows that these molecules have a high positive correlation in the immune response set (Figure 3B).

### 3.3. Expression Profiles of Differentially Expressed cicrRNAs

Meanwhile, we predicted new circRNAs. There were 80 new circRNAs differentially expressed in the AvsB group, 74 new circRNAs differentially expressed in the AvsC group, and 63 new circRNAs in the BvsC group that were significantly changed (Figure 4A–C). The results of the GO analysis showed that the target genes of the AvsB group were enriched in the negative regulation of cell adhesion, negative regulation of cell development, and description processes, which were mostly related to the negative regulation of biological processes (Figure 4D). The AvsC group was related to processes such as the endomembrane system, microtubule motor activity, and GMP biosynthetic process (Figure 4E). The target genes of the BvsC group were mainly involved in pathogenesis and enzymatic activity (Figure 4F). Most of the target genes in the AvsB group were related to material metabolism pathways (Figure 4G). The AvsC group was significantly enriched in protein export, the cGMP−PKG signaling pathway, and basal transcription factors pathways (Figure 4H). The target genes of the BvsC group were mainly enriched in the p53 signaling pathway, cell cycle, and mismatch repair pathway (Figure 4I).

The P set had 60 intersection circRNAs. There were 46 circRNAs in the intersection of the I–V set (Appendix A). These predicted new circRNAs were involved in different molecular events during SADS−CoV infection, which can provide references for follow−up studies. The HAllA association analysis was performed on these circRNAs and mRNAs. In the P set, novel_circ_0006647, novel_circ_0009773, novel_circ_0007348 and novel_circ_0008394 all had the highest association with ROCK2 (Rho associated coiled−coil−containing protein kinase 2) (https://www.genecards.org/Search/Keyword?queryString=ROCK2 accessed on 27 January 2023) and were positively correlated (Figure 5A). In the I–V set, the highest correlation was between novel_circ_0002913 and PSMB9 (Proteasome 20S subunit beta 9) (https://www.genecards.org/Search/Keyword?queryString=PSMB9 accessed on 27 January 2023), CD3D (CD3 delta subunit of T cell receptor complex) (https://www.genecards.org/Search/Keyword?queryString=CD3D), CACYBP (calcyclin−binding protein) (https://www.genecards.org/Search/Keyword?queryString=CACYBP accessed on 27 January 2023), CD3G (CD3 gamma subunit of T cell receptor complex) (https://www.genecards.org/Search/Keyword?queryString=CD3G accessed on 27 January 2023), and there was a significant negative correlation between them (Figure 5B). A notable phenomenon is that most of the top circRNAs and mRNAs in the I–V set have negative correlations (Figure 5C). The results of the association analysis between circRNAs and miRNAs showed that in the I–V set, circRNA novel_circ_0009547 had the highest correlation with ssc−miR−1388, which were negatively correlated. In the P set, miRNA novel_424 (new prediction) had the highest correlation and a positive correlation with circRNA novel_circ_0000365, novel_circ_0005562, and novel_circ_0012577. Secondly, the miRNAs ssc−miR−504 (https://www.mirbase.org/textsearch.shtml?q=ssc−miR−504 accessed on 27 January 2023) and ssc−miR−574−3p (https://www.mirbase.org/textsearch.shtml?q=ssc−miR−574−3p accessed on 27 January 2023) were highly correlated with the circRNA novel_circ_0000931, implying a high positive correlation between these molecules in the P set (Figure 5D).

### 3.4. Expression Profiles of Differentially Expressed lncRNAs

There were 132 differentially expressed lncRNAs in the AvsB group, 51 differentially expressed lncRNAs in the AvsC group, and 146 lncRNAs in the BvsC group with significant changes (Figure 6A–C). More lncRNAs are involved in immune response regulation compared to the low pathogenic group. The GO results of the three groups were similar, and most of them were related to protein binding and metabolism (Figure 6D–F). The KEGG analysis showed that lncRNAs in the AvsB group were significantly enriched in Fc gamma R−mediated phagocytosis, mTOR signaling, and some metabolism−related pathways (Figure 6G). In the AvsC group, lncRNAs were significantly enriched in T cell receptor signaling, p53 signaling, and Jak−STAT signaling pathways (Figure 6H). In the BvsC group, lncRNAs were enriched in Fc gamma R−mediated phagocytosis, the T cell receptor signaling pathway, phagosomes, and other pathways (Figure 6I). Our results also suggest that differentially expressed lncRNAs can be involved in regulating multiple pathways, such as immune responses, cellular metabolism, and thus the body’s response to viruses. In addition, we found that the differentially expressed lncRNAs of the three groups were significantly enriched in the inflammatory bowel disease pathway, which may be related to the ability of SADS−CoV to cause diarrhea and intestinal damage.

There are seven intersection lncRNAs in the P set, and one intersection lncRNA in the I–V set, which is related to the host’s immune response to SADS−CoV (Appendix A). Then we analyzed the association between lncRNA and miRNA by the HAllA association method, and the results showed that in the P set, LNC_008160 was significantly positively correlated with ssc−miR−10391 and novel_424. LNC_011487 was significantly negatively correlated with ssc−miR−215, and further studies are needed on whether there is a competitive endogenous RNA relationship between the two genes (Figure 7A). The association analysis between lncRNA and mRNA showed that LNC_011487 was positively associated with PI3 and negatively associated with SLC37A3 in the P set (Figure 7B). The above results indicate that differentially expressed lncRNAs can play a role in the molecular events related to the potential pathogenicity, lethality, and inflammatory response of secondary bacterial infections in combination with miRNAs and mRNAs.

### 3.5. Expression Profiles of Differentially Expressed TUCP

Transcripts of uncertain coding potential (TUCP) are part of long noncoding RNAs, which include short open reading frames and can be translated into small peptides [22]. After analyzing the differentially expressed TUCPs, it was found that there were 66 differentially expressed TUCPs in the AvsB group, 25 differentially expressed TUCPs in the AvsC group, and 92 TUCPs in the BvsC group with significant changes (Figure 8A–C). The results showed that there were more differentially expressed TUCPs involved in the regulation of immune response. After the GO enrichment analysis of TUCP co−expressed genes, it was found that the TUCP in the AvsB group was mainly related to RNA binding and viral genome replication, indicating that the differential expression of TUPC in the AvsB group was related to pathogenicity (Figure 8D). The TUCP in the AvsC group was mainly related to inflammatory response, the viral envelope, viral membrane, and other processes, indicating that the differential expression of TUCP in this group indeed corresponds to the basis of the grouping and is related to pathogenicity and immune response (Figure 8E). In the BvsC group, the TUCP was associated with binding and the proteasome complex (Figure 8F). The KEGG enrichment analysis of TUCP co−expressed genes showed that TUCP was significantly enriched in p53 signaling, RNA transport, mTOR signaling, and other pathways in the AvsB group (Figure 8G). The AvsC group was significantly enriched for immune−related regulation, such as T cell receptor signaling, primary immunodeficiency, and NF−kappa B signaling (Figure 8H). Like the AvsB group, the BvsC group was also significantly enriched in immune−related pathways, indicating that the differential expression of TUCP between the AvsC and BvsC groups was more involved in the body’s immune regulation (Figure 8I). 

The results of the HAllA association analysis showed that in the P set, TUCP_002240 had the strongest association with LYZ and TSPAN1, and was positively correlated (Figure 9A). In the I–V set, TUCP_03109, TUCP_000239, TUCP_003235, TUCP_003225, and TUCP_003878 were all significantly and positively correlated with multiple mRNAs (Figure 9B). This shows that these TUCPs can play a role in association with mRNA.

### 3.6. Combined Analysis of mRNAs, miRNAs, lncRNAs, TUCP, cicrRNAs

We performed a joint analysis of the interaction between differentially expressed RNAs. In the I–V set, 67 interrelated differentially expressed RNAs were obtained to construct an association network. The network consists of 67 nodes (28 mRNAs, 32 circRNAs, 3 miRNAs, and 4 TUCPs) and 93 edges (Figure 10A). In the P collection, 134 interrelated differentially expressed RNAs were obtained to construct an association network. The network consists of 134 nodes (53 mRNAs, 57 circRNAs, 4 lncRNAs, 17 miRNAs, and 3 TUCPs) and 327 edges (Figure 10B).

## 4. Discussion

To our knowledge, this study is the first to comprehensively detect differentially expressed RNAs by RNA−Seq, including from three subgroups (virulent strain group vs control group, virulent strain group vs attenuated strain group, and attenuated strain group vs control group) of mRNA, lncRNA and miRNA, and cicrRNA and TUCP. Based on these differentially expressed RNAs, we performed an intersection analysis of molecular events related to pathogenicity and immune response and analyzed the associations between differentially expressed RNAs and mRNA, lncRNA, miRNA, cicrRNA, and TUCP by a HAllA association. The differences between the virulent strain and the control group, the virulent strain and the attenuated strain, and the attenuated strain and the control group were compared for each RNA. We also compared the intersection of AvsC and AvsB for the differentially expressed RNAs, denoted as set P, and the intersection of BvsC and AvsC, denoted as set I–V. We conducted an in−depth analysis of the associations between RNAs (circRNA−miRNA, circRNA−mRNA, mRNA−miRNA, TUCP−mRNA, lncRNA−miRNA, and lncRNA−mRNA) from two dimensions of set P and set I–V through a HALLA association.

From the analysis of differential mRNAs in the P set, we found that TGM3 was associated with virus entry into host cells and the virus cycle. In our setting, the P set represents the molecular events related to the potential pathogenicity, lethality, and inflammatory response of the secondary bacterial infection, so TGM3 can be used as the representative mRNA of the P set. Studies have shown that among the differential miRNAs, ssc−miR−1285 deserves attention, although the existing research results have not indicated its function [23]. However, ssc−miR−1285 was significantly positively correlated with S100G in the correlation analysis of miRNA and mRNA HALLA in the P set in this study. In the absence of IL−10, S100G expression was elevated, inflammatory changes were reduced after the induction of colitis, and S100G inhibited monocyte chemoattractant protein−1 (MCP−1) production by inhibiting NF−κB activation, which is an important anti−inflammatory mediator in fibroblasts after the induction of colitis [24]. Whether ssc−miR−1285 can interact with S100G in terms of anti−inflammatory response still needs to be explored in further research.

In the association analysis of differentially expressed cicrRNA, it was found that the mRNA with the highest correlation with cicrRNA was ROCK2. ROCK2 (Rho−related coiled−coil−containing protein kinase−2) is a protein with multiple functions. Its activity is up−regulated in acute inflammatory injury and chronic diseases, such as diabetes, metabolic syndrome and idiopathic pulmonary fibrosis. The disruption of ROCK2 functions to restore homeostasis by inhibiting pro−inflammatory TH17 cells and promoting regulatory T cells [25]. The miRNA with the highest association with cicrRNA is the newly predicted miRNA novel_424, and the association with multiple cicrRNAs is the highest. The differentially expressed lncRNA LNC_008160 is also significantly associated with novel_424. In the P collection, LNC_011487 was significantly negatively correlated with ssc−miR−215, and LNC_011487 was positively correlated with PI3. The specific role of LNC_011487−ssc−miR−215−PI3 needs further study. The mRNAs most associated with TUCP_002240 are LYZ and TSPAN1; LYZ is a component of innate immunity, and TSPAN1 has a regulatory role in cancer [26,27]. These results indicate that there are multiple molecular mechanisms that regulate the potential pathogenicity, lethality of the virus, and inflammatory response to secondary bacterial infection. This study has contributed to a better understanding of the inflammatory response to pathogenic, lethal, and secondary bacterial infections.

FUT2, CCL5, SLA−DQA1, and CXCL9 in the differential mRNA of the I–V intersection were all related to immune response. This could be explained by the fact that the intersection is associated with the host immune response to SADS−CoV. The differentially expressed miRNA ssc−miR−1285 is associated with FUT2, an enzyme known to be responsible for the addition of fucose to proteins or lipids through α−1,2−fucosylation on the intestinal mucosa, which can act as an attachment site and carbon source for gut bacteria [28]. Tong et al. analyzed the gut microbiota of healthy subjects and they found altered gut microbiota in Fut2 non−secretors [29]. Interestingly, previous studies have shown that in patients with primary sclerosing cholangitis, Fut2 non−secretors exhibit a different composition of bile microbiota compared to that of Fut2 secreters [30]. These findings suggest a potential regulatory role of Fut2 on the gut microbiota. So far, no one has investigated the regulatory effect of virulent and attenuated SADS−CoV strains on the intestinal flora, and it is worth considering whether FUT2 plays a role in this. The functions of most differentially expressed TUPCs are still unknown, but their functions can be mined through their associated RNAs. Among the differentially expressed TUPC−associated mRNAs in the I–V set, PSMB9 is an immunoproteasome component [31]. CCL5 induces the in vitro migration and recruitment of T cells, dendritic cells, eosinophils, NK cells, mast cells, and basophils [32]. Functional polymorphisms in the regulatory regions of the VNN1 gene are associated with susceptibility to inflammatory bowel disease [33]. The above results indicate that we can focus on some highly related RNAs in the I–V collection, and the regulation between them may be very helpful for the study of immune and pathogenic mechanisms. 

The GO and KEGG analyses of differentially expressed RNAs in the AvsB group found that most RNAs were involved in protein binding, material metabolism, and autophagy [34]. Among them, autophagy is a key cellular response to pathogen infection. Studies have shown that the process of autophagy regulates intestinal microbiota, and an increase in microbiota is associated with diarrhea [35,36]. In the study of our group, it was found that the structure of gut microbiota changed significantly after SADS−CoV infection in piglets (data unpublished). At the same time, autophagy has also been identified to play a role in the immune system, participating in viral suppression or enhancement processes [37]. The differentially expressed RNAs of AvsB were mainly enriched in Fc gamma R−mediated phagocytosis, Fc epsilon RI signaling pathway, mTOR signaling, and some metabolism−related pathways. Phagocytosis is a key event in the immune system, allowing cells to engulf and eliminate pathogens, and Fc gamma receptors can mediate phagocytosis, suggesting that differentially expressed RNAs may be indirectly involved in the regulation of the immune system [38,39]. Zhang et al. showed by comprehensive functional analysis of SADS−CoV on host mRNA profiles that SADS−CoV infection induced strong immune responses, including innate immunity and cytokine–cytokine receptor interactions, and similar findings were indeed found in our study [40]. The analysis of differentially expressed RNAs of AvsC showed that differentially expressed RNAs were mainly related to immune response, catalysis, metabolism, and other processes, and were enriched in disease−related pathways, antigen processing and presentation, T cell receptor signaling, p53 signaling, Jak−STAT signaling, primary immunodeficiency, NF−kappa B signaling, and other pathways. The differential mRNAs in the AvsC group are involved in more immune and disease−related signaling pathways, which are basically consistent with the molecular events expected by our grouping. The differentially expressed RNA GO enrichment term in the BvsC group was a little different from the other groups, and most of them were related to metabolism and immunity. Toll−like receptor signaling, NF−kappa B signaling, T cell receptor signaling, and other pathways are found in KEGG enrichment. Like the AvsB group, the BvsC group was also significantly enriched in the Fc gamma R−mediated phagocytosis and Fc epsilon RI signaling pathways. In addition, the BvsC group had the most significant enrichment of differentially expressed miRNAs in the intestinal immune pathway network for IgA production, indicating that infection with attenuated strains stimulates intestinal immunity [41]. 

After the association analysis of the differentially expressed RNAs, a transcriptome−wide association network graph was constructed, and in the P set, the core differentially expressed RNA was OGDH, which was significantly associated with 20 differentially expressed RNAs. In the pathological state of sepsis, OGDH amplifies the inflammatory response through the MAPK pathway, releases pro−inflammatory factors, and induces acute lung injury [42]. In the I–V set, the core differentially expressed RNA is novel_circ_0001750, which is significantly associated with 13 differentially expressed RNAs, including 12 mRNAs and 1 miRNA, but it has not been studied in detail yet. The association network diagram facilitates our study of the role of association RNAs during SADS−CoV infection.

Of course, our research has certain limitations. First, due to the influence of African swine fever, experimental animals are rare and valuable. We only performed three biological replicates during sampling, and more replicates should be performed to ensure the reliability of the experiment. Secondly, we currently only predict the function of differentially expressed RNA, which we will further study in the future.

## Figures and Tables

**Figure 1 microorganisms-11-00409-f001:**
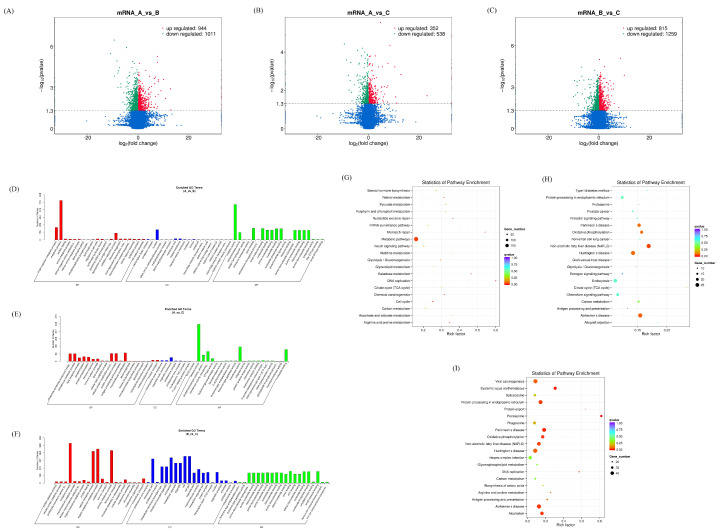
Expression profiles of differentially expressed mRNAs. (**A**–**C**) Volcano plot of differentially expressed mRNAs in AvsB group (**A**), AvsC group (**B**), and BvsC group (**C**). (**D**,**F**) GO analysis of differentially expressed mRNAs in AvsB group (**D**), AvsC group (**E**), and BvsC group (**F**). (**G**,**I**) KEGG pathway analysis of differentially expressed mRNAs in AvsB group (**G**), AvsC group (**H**), and BvsC group (**I**).

**Figure 2 microorganisms-11-00409-f002:**
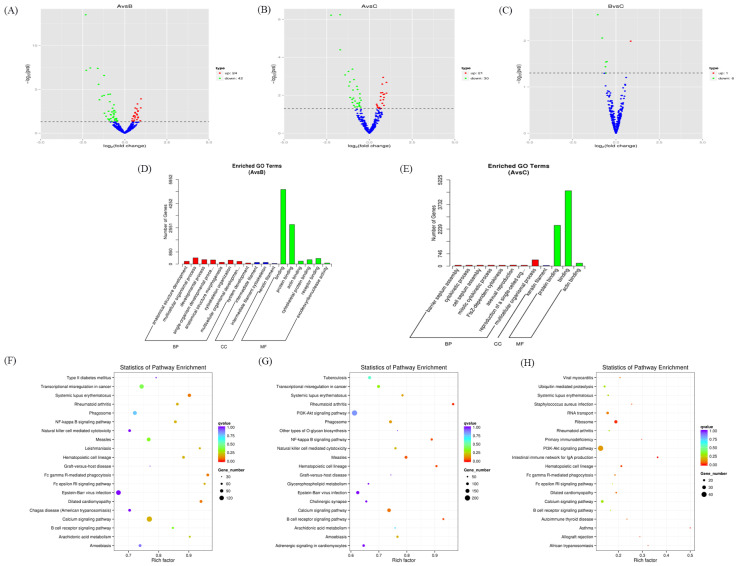
Expression profiles of differentially expressed miRNAs. (**A**–**C**) Volcano plot of differentially expressed miRNAs in AvsB group (**A**), AvsC group (**B**), and BvsC group (**C**). (**D**,**E**) GO analysis for neighbor gene functions of miRNA in AvsB group (**D**) and AvsC group (**E**). (**F**–**H**) KEGG annotation for neighbor gene functions of miRNA in AvsB group (**F**), AvsC group (**G**), and BvsC group (**H**).

**Figure 3 microorganisms-11-00409-f003:**
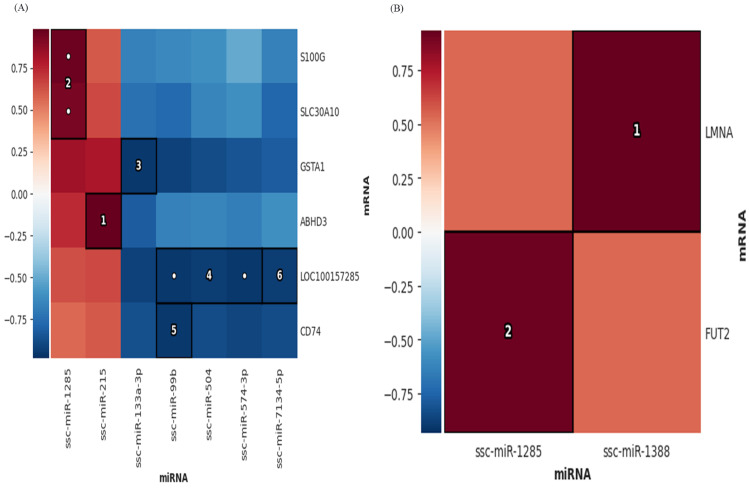
Association analysis of miRNAs and mRNAs. (**A**) Association results between miRNA and mRNA in the P set. (**B**) Association results between miRNA and mRNA in the I−V set. Color bars represent correlation coefficients: red for positive correlation and blue for negative correlation.

**Figure 4 microorganisms-11-00409-f004:**
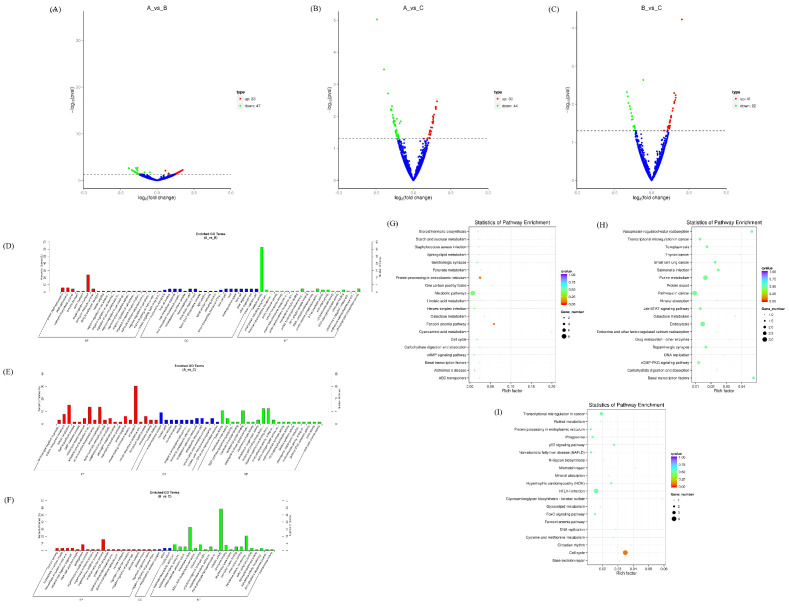
Expression profiles of differentially expressed cicrRNAs. (**A**–**C**) Volcano plot of differentially expressed cicrRNAs in AvsB group (**A**), AvsC group (**B**), and BvsC group (**C**). (**D**–**F**) GO analysis for neighbor gene functions of cicrRNAs in AvsB group (**D**), AvsC group (**E**), and BvsC group (**F**). (**G**–**I**) KEGG annotation for neighbor gene functions of cicrRNAs in AvsB group (**G**), AvsC group (**H**), and BvsC group (**I**).

**Figure 5 microorganisms-11-00409-f005:**
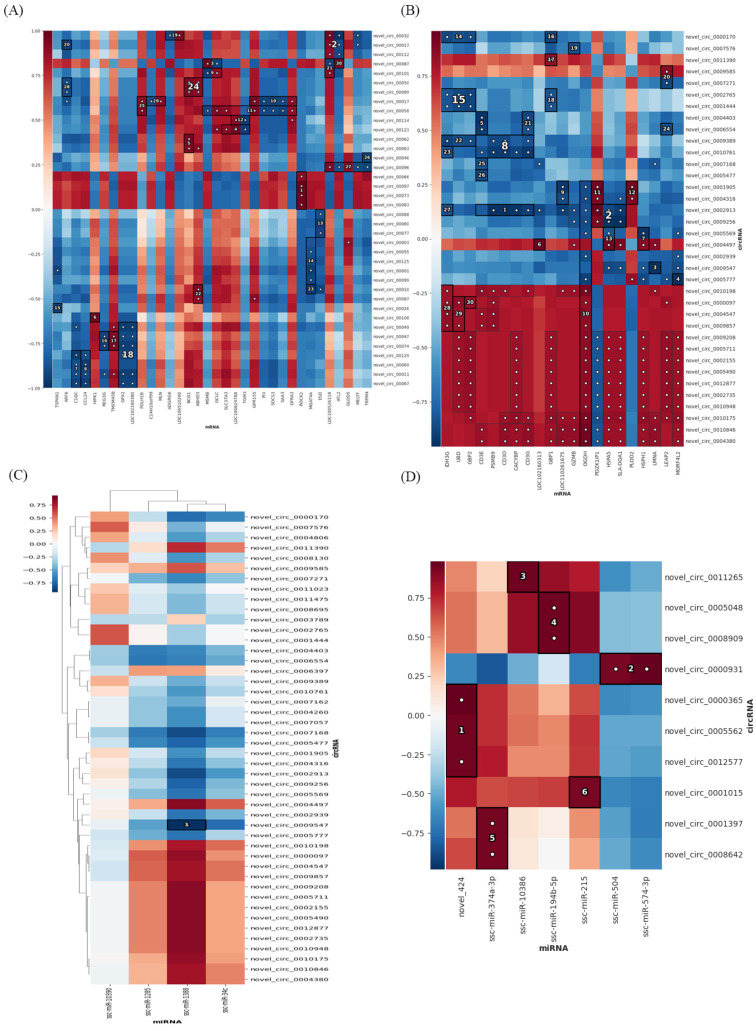
Association analysis of cicrRNAs−mRNAs and cicrRNAs−miRNAs. (**A**) Association results between cicrRNAs and mRNA in the P set. (**B**) Association results between cicrRNAs and mRNA in the I−V set. (**C**) Association results between cicrRNAs and miRNA in the I−V set. (**D**) Association results between cicrRNAs and miRNA in the P set. Color bars represent correlation coefficients: red for positive correlation and blue for negative correlation.

**Figure 6 microorganisms-11-00409-f006:**
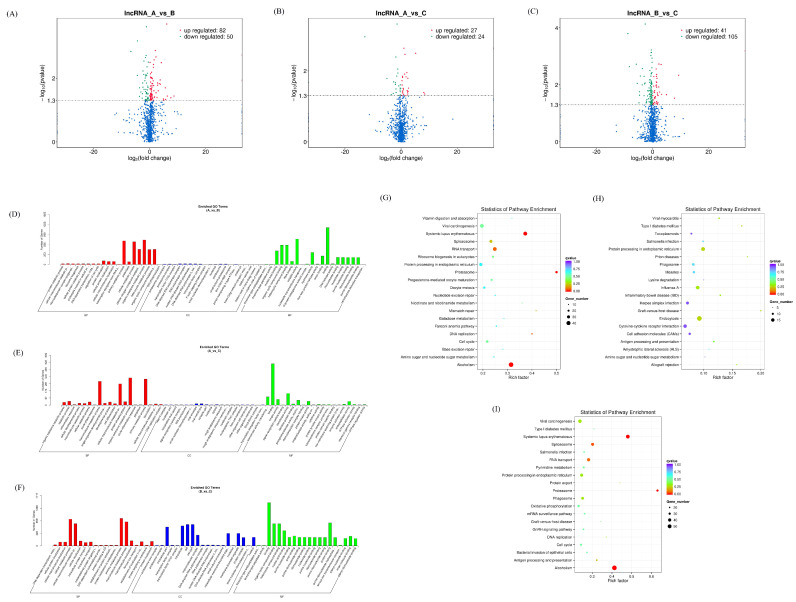
Expression profiles of differentially expressed lncRNAs. (**A**–**C**) Volcano plot of differentially expressed lncRNAs in AvsB group (**A**), AvsC group (**B**), and BvsC group (**C**). (**D**–**F**) GO analysis for neighbor gene functions of lncRNAs in AvsB group (**D**), AvsC group (**E**), and BvsC group (**F**). (**G**–**I**) KEGG annotation for neighbor gene functions of lncRNAs in AvsB group (**G**), AvsC group (**H**), and BvsC group (**I**).

**Figure 7 microorganisms-11-00409-f007:**
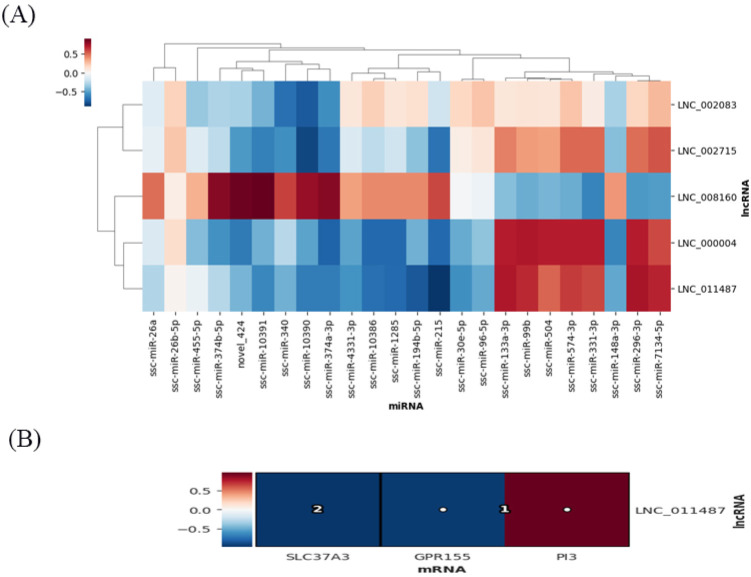
Association analysis of lncRNAs−miRNAs and lncRNAs−mRNAs. (**A**) Association results between lncRNAs and miRNA in the P set. (**B**) Association results between lncRNAs and mRNA in the P set. Color bars represent correlation coefficients: red for positive correlation and blue for negative correlation.

**Figure 8 microorganisms-11-00409-f008:**
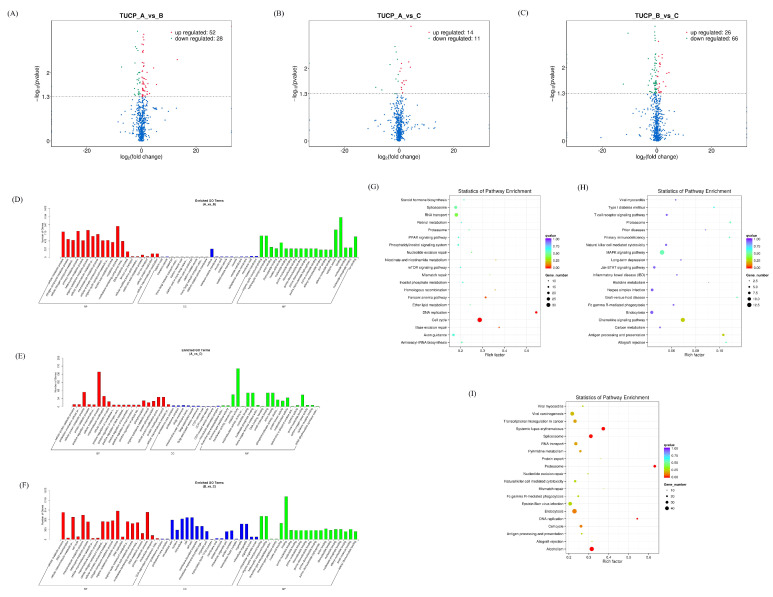
Expression profiles of differentially expressed TUCP. (**A**–**C**) Volcano plot of differentially expressed cicrRNAs in AvsB group (**A**), AvsC group (**B**), and BvsC group (**C**). (**D**–**F**) GO analysis for neighbor gene functions of TUCP in AvsB group (**D**), AvsC group (**E**), and BvsC group (**F**). (**G**–**I**) KEGG annotation for neighbor gene functions of TUCP in AvsB group (**G**), AvsC group (**H**), and BvsC group (**I**). AvsC and AvsB have three intersection TUCPs, and BvsC and AvsC have five intersection TUCPs (Appendix A).

**Figure 9 microorganisms-11-00409-f009:**
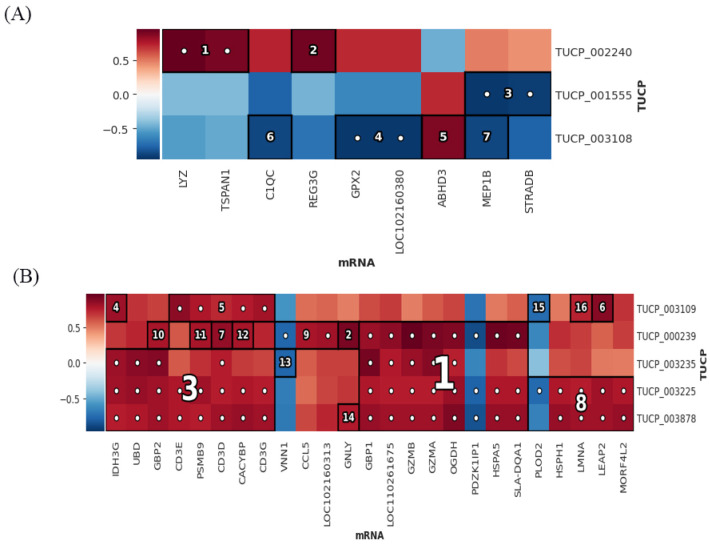
Association analysis of TUCP and mRNAs. (**A**) Association results between TUCP and mRNAs in the P set. (**B**) Association results between TUCP and mRNAs in the I–V set. Color bars represent correlation coefficients: red for positive correlation and blue for negative correlation.

**Figure 10 microorganisms-11-00409-f010:**
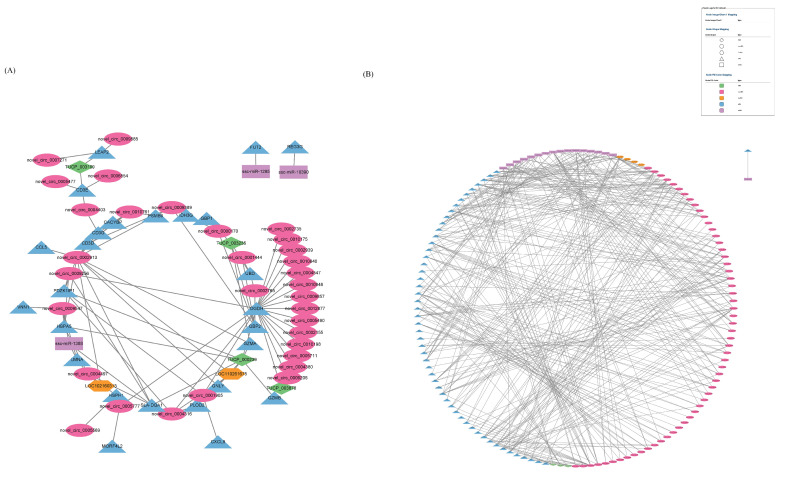
Combined analysis of mRNAs, miRNAs, lncRNAs, TUCP, and cicrRNAs. (**A**) The network of I–V set. (**B**) The network of P set.

## Data Availability

The original contributions presented in the study are included in the article. Further inquiries can be directed to the corresponding authors.

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
