# Peer review of "Combined Analysis of the Whole Transcriptome of Piglets Infected with SADS−CoV Virulent and Avirulent Strains"

_microorganisms, 2023, doi:10.3390/microorganisms11020409_

Round 1
Reviewer 1 Report
In this study, Li et al., analyzed and compared the whole transcriptome including mRNA, miRNA, lncRNA, cicrRNA and TUCP of the virulent and avirulent SADS-CoV strains infected piglet intestinal tissues. Relevant analysis has certain significance for understanding the pathogenesis of the virus.
Main comments:
1. The manuscript is only analyzed by transcription, predict the function of differentially expressed RNA. It is necessary to add the functional verification of important changed RNA and increase the overall quality of the manuscript.
2. The data of animal experiments are not reflected in the manuscript at all, and the relevant data are not published. The author needs to consider the addition of some animal experimental data or the citation of relevant data after publication.
Minor comments:
1. The first sentence in the abstract is missing the virus.
2. Line 90, the names of the other viruses appeared for the first time, requiring full names.
3. Lines 92-94, the statement is in error and needs modification. The virulent and avirulent.
4. The language of the manuscript is subject to multiple errors, requiring careful proofreading and revision of the entire article.
Author Response
Point by point responses to the reviewer:
First, we would like to thank the reviewers and the editor for the positive and constructive comments and suggestions.
Main comments:
- The manuscript is only analyzed by transcription, predict the function of differentially expressed RNA. It is necessary to add the functional verification of important changed RNA and increase the overall quality of the manuscript.
Answers: In this article we focus on the analysis, the experimental verification will be detailed in the next article
- The data of animal experiments are not reflected in the manuscript at all, and the relevant data are not published. The author needs to consider the addition of some animal experimental data or the citation of relevant data after publication.
Answers: We have followed your suggestion to cite the relevant studies already available in this experimental group. The title of this article is Attenuation of a virulent swine acute diarrhea syndrome coronavirus strain via cell culture passage. Line 73-75.
Minor comments:
- The first sentence in the abstract is missing the virus.
Answers: We have made corrections based on your suggestions.
- Line 90, the names of the other viruses appeared for the first time, requiring full names.
Answers: We have made corrections based on your suggestions. Line 107-109
- Lines 92-94, the statement is in error and needs modification. The virulent and avirulent.
Answers: We have made corrections based on your suggestions. Line 112-113
- The language of the manuscript is subject to multiple errors, requiring careful proofreading and revision of the entire article.
Answers: We apologize for the poor language of our manuscript. We have now worked on both language and readability and have also involed native english speaker for language corrections. We really hope that the flow and language level have been substantially improved.

Reviewer 2 Report
No comments.
Author Response
Point by point responses to the reviewer:
First, we would like to thank the reviewers and the editor for the positive and constructive comments and suggestions. Please find my itemized responses in below and my revisions in the re-submitted files.
Reviewer 3 Report
Please see attached review document

Author Response
Point by point responses to the reviewer:
First, we would like to thank the reviewers and the editor for the positive and constructive comments and suggestions. Please find my itemized responses in below and my revisions in the re-submitted files.
Main comments:
Not enough citations to pertinent literature in introduction, materials,and discussion. Descriptions of analysis lacking to properly vet the experimental procedures.
Thank you for your valuable comments, we have improved and revised these sections based on your suggestions.
- Line 35 and 37 “porcine acute diarrhea syndrome”should read “swine acute diarrhea syndrome” to be in congruence with the acronym (SADS).
Answers: We have changed lines 35 and 37 from "Acute diarrhea syndrome in pigs" to "Acute diarrhea syndrome in pigs" based on your suggestion.
- Line 40 “coated RNA virus”do the authors mean enveloped?
Answers: We are sorry for this error, we have changed "coated RNA virus" to Envelope virus.
- Line 56-58 may want to cite Edwards et al (https://doi.org/10.1073/pnas.2001046117) or other pertinent literature pertaining to the current SADS-CoV-2 receptor search.
Answers: Based on your suggestions, we have cited the relevant literature.
- “The passaged strains SADS-CoV-CN/GDWT/2017-P7 and SADS-CoV-CN/GDWT/2017-P83 of the SADS-CoV-CN/GDWT/2017 strain were isolated and preserved by our laboratory”Please describe how the viruses were passaged, numbers of passages in which cell types. Were specific measures taken to reduce the virulenceofstrains?
Answers: We have revised the text according to your suggestions.
Serum-free DMEM containing 10 μg/ml trypsin (Gibco). Infected cells were incubated at 37°C in an incubator containing 5% CO2 and monitored daily for cytopathic effects (CPE). When 90% of cells show CPE, harvest cells, freeze and thaw three times, and centrifuge at 8000 x g for 10 min. The supernatant is collected and stored at -80°C as a stock solution for the next passaging. Using the same approach, 82 subsequent passages were performed in Vero cells. Purify every 10 generations, starting from the 8th generation of cell culture. Viral titers were determined by TCID50 every 5 or 10 generations. Line 87-98.
- More details regarding intestinal harvesting are necessary. Which intestines, large or small How much of the intestines were harvested normalized by weight or other? How were the intestines lysed, specifically what was in the lysis solution?
Answers: Thank you for your suggestion and we have modified the manuscript accordingly. The intestinal tissue taken in our experiments was the ileum of the small intestine tissue, 0.2 g per sample. intestinal tissues were lysed by grinding after adding TRIZOL reagent (Invitrogen, USA). Line 107-118.
- Were any of the differentially expressed genes confirmed utilizing traditional methodssuch as quantitative PCR?
Answers:Thank you very much for your advice. This part has been validated in our other experiments. In view of the insufficient experimental samples, we did not do the validation in this article. The data of this experiment shows that the difference within the three biological replicate groups is very small, and the data are quite reliable even without validation. We will complete data validation and perform functional validation if samples are available at a later stage.
- Library Construction and Sequencing Process-Please cite existing pertinent literature for library construction techniques. Should probably be at least 4-5 additional citations for each step of this process.
Answers: We have enriched our references based on your suggestions.
- Line 66-68 age. “We used virulent and avirulent strains for animal challenge experiments, and clinical symptoms showed that tissue lesions, anal swab detoxification, and tissue tropism were relatively weakened (data not published).” Sentenceneeds clarification, as written seems like both virulent an avirulent strains are weakened?
Answers: We apologize for the lack of clarity in our description, only avirulent strains are weakened here, and we have made corrections in the manuscript. Line 72-73
- What are the genetics and background of the pigs used in the study?
Answers: The pig we use is a cross between landrace as the mother and Tang pig as the father, the resulting Two-way crossbred as the father and then crossed with Duroc as the mother to produce a three-way crossbred.
- Authors appear to neglect previous studies on RNAseq with SAD-CoV in vitro. Could the authors compare their in vivo data to that of Zeng etal 2020 https://doi.org/10.3389/fvets.2020.00492 who used IPEC-J2 cells with RNAseq briefly comparing their findings? There is also a publication by https://www.ncbi.nlm.nih.gov/pmc/articles/PMC8070899/in which they utilized vero cellsRNAseq (I am not as much a fan of this study due to the obvious defects in innate immune signaling in Vero cells.).
Answers: Zhang et al. showed by comprehensive functional analysis of SADS-CoV on host mRNA profiles that SADS-CoV infection induced strong immune responses, including innate immunity and cytokine-cytokine receptor interactions, and similar findings were indeed found in our study, but our study focused more on the differences that existed in comparing the effects of virulent and avirulent strains on the organism and was analyzed in depth by both pooled P set and I-V set dimensions.
It seems to me that in the discussion of the study by Zeng et al. they emphasize that the immune escape mechanism causes downregulation of host genes, but in SADS-CoV infection itself suppresses interferon production, and I wonder if this contradicts the study by Zeng et al. and may require a lot of experiments to be performed to prove it.
- Many of the figure axis labels(1, 2,4,6,8, 10)are too small to be read even with zooming in on the provided .pdf file.
Answers: We have made changes based on your comments. In addition, we can also provide vector graphics
- The raw data files should be shared and referenced on an open access server such as the Gene Expression Omnibus https://www.ncbi.nlm.nih.gov/geo/. It is frustrating for other bioinformatacists to read an interestingarticle and then not be able to further explorethis important data.
Answers: We will then share the raw data files on an open server based on your suggestions.
- “the piglets were sent to the animal room when they were 5days old” Please clarify whether pigs were housed separately to ensure no cross infection occurred.
Answers: Our piglets are housed in separate animal rooms to ensure no cross. We have made changes in the manuscript. Line 104.
- “During the peak incidence period from day 3 to day 5 post-challenge, three piglets were randomly selected to be bled and slaughtered,” Three pigs per group or three total pigs?
Answers: Our experiment was to randomly select three piglets per group for dissection, which we have modified in the manuscript. Line113-114.
- More details needed regarding the “Small RNA sample pre kit”
Answers: We have described in detail in the manuscript according to your suggestions.
Small RNA Sample Preparation Kit is designed to generate small RNA libraries directly from total RNA. Line 122-124.
- “lively piglets with basically the same weight, age and age were selected” –specific details are needed.
Answers: Thank you for your suggestions, we have revised them in the manuscript. We selected piglets weighing around two kilograms, responsive, energetic, with high appetite and sensitive to external stimuli for the experiment. Line 100-102.
- Was virulence of A, B, and C strains confirmed during this trial in parallel?
Answers: The attacking amount was 106.3 TCID50/mL, 12mL for both groups A and B, and 12mL DMEM for group C. Line 109-113
- Hierarchical All-against-All association (HAllA) is a computational method for finding multi-resolution associations in high-dimensional, heterogeneous datasets, and for discovering significant relationships between data features with high power (Citation).
Answers: Thank you for your suggestion, we have cited the relevant literature.
- Define AvsC and AvsB as group A versus group B, etc. This can be initially confusing to the reader.
Answers: Thank you for your suggestions, we have made changes in the manuscript. Line 26-27.
- Further explanation for defining AvsC and AvsB as “represents molecular events associated with viral potential pathogenicity, lethality, and inflammatory responses secondary to bacterial infection, P set” while BvsC and AvsC are attributable to host immune response may be needed.
Answers: We use A, B, and C to represent the lethality group, the low pathogenicity, and the control group. the molecular events associated with pathogenicity in A-vs-C are consistent with those in A-vs-B. the immune response generated in A vs C contains a response to bacterial infection secondary to SADS-CoV, and the immune response generated in B vs C is purely a host immune response to SADS-CoV. Systematic errors were randomized. Line181-186.
- Results-- When describing individual genes it would be good to provide a reference to the gene function either a uniport designation or citation or primary literature. At minimum define the gene name acronym. For example NIPBL (Nipped-B-Like Proteinor NIPBL Cohesin Loading Factor)(https://www.genecards.org/cgi-bin/carddisp.pl?gene=NIPBL). This could direct the reader to more in-depth information regarding the gene function.
Answers: Thank you for your suggestions, we have made changes in the manuscript.
- While a gene like NIPBLis found in AvsB comparisons, how is the same gene altered in AvsC and BvsC? This type of 3-way comparison might be more telling of potential contributions to disease severity.
Answers: Thank you for your suggestion, we have made a detailed comparison in the manuscript based on your suggestion.
- The researchers may wish to consider including more reason ing for HAllA multi-omics association. What is the rationale for looking at crcRNA vs mRNA, circRNA vs miRNA, etc. Is there previous research that suggests these types of correlations are biologically relevant? This existing data should be cited.
Answers: HAllA (Hierarchical All-against-All association) is a computational method for finding multi-resolution associations in high-dimensional, heterogeneous data sets. It is used to discover significant relationships between data features with high power. It is robust to data types (robustness), operates on both continuous and categorical values, and works well on both homogeneous datasets and heterogeneous data.
In biology, mRNA/circRNA, miRNA/mRNA, lncRNA/mRNA, these RNA pairs may play some important roles as reciprocal ceRNAs. We believe that HALLA can better represent the association between them
- Define P set in line 162 as this is the first time the term is encountered.
Answers: Thank you for your suggestions, we have made changes in the manuscript. Line81.

Round 2
Reviewer 1 Report
there is no comment.